# Spatiotemporal Neural Diffeomorphic Flow for Cardiac Atlas and Motion Recovery

**Yiyang Xu**[1]                                                      YIYANG.1.XU@KCL.AC.UK
[1] *School of Biomedical Engineering and Imaging Sciences, King's College London, UK*
**Esther Puyol-Antón**[2]                                            EPUYOLANTON@HEARTFLOW.COM
**Matthew Sinclair**[2]                                              MSINCLAIR@HEARTFLOW.COM
[2] *Heartflow Inc., Mountain View, USA*

**Amedeo Chiribiri**[1]                                              AMEDEO.CHIRIBIRI@KCL.AC.UK
**Steven A Niederer**[3]                                             STEVEN.NIEDERER@IMPERIAL.AC.UK
[3] *National Heart and Lung Institute, Imperial College London, UK*

**Alistair A Young**[1]                                              ALISTAIR.YOUNG@KCL.AC.UK

## Abstract

Spatiotemporal cardiac atlases built from cardiovascular magnetic resonance (CMR) data enable population-level analysis of cardiac anatomy and motion, yet existing methods lack topological guarantees and dense inter-subject correspondences. We extend Neural Diffeomorphic Flow (NDF) to the spatiotemporal domain, constructing a population-based left ventricular atlas with dense, topology-preserving correspondences across individuals and time frames. Evaluated on UK Biobank CMR data, our method achieves 1.4 mm chamfer distance for motion recovery of the full cardiac cycle from a single end-diastolic frame.

**Keywords:** Cardiovascular Magnetic Resonance, Spatiotemporal Cardiac Atlases, Motion Recovery

## 1. Introduction

Cardiovascular diseases remain the leading cause of mortality worldwide (Mendis et al., 2011), driving demand for computational approaches that quantify cardiac structure and dynamics non-invasively. Cardiovascular magnetic resonance (CMR) is the gold standard for capturing full-cycle cine sequences at high spatiotemporal resolution without ionising radiation (Petersen et al., 2016). Computed tomography (CT) offers finer spatial detail but is limited to a single cardiac phase due to dose constraints. Spatiotemporal cardiac atlases establish shared reference spaces for cross-subject comparison and disease characterisation (Young and Frangi, 2009), yet classical approaches (Bai et al., 2015) require costly registration and fixed discretisations, while methods such as CHeart (Qiao et al., 2024) and CardiacFlow (Ma et al., 2025) learn implicit atlases via latent spaces but lack topological guarantees and the ability to map between atlas and individual. Neural Diffeomorphic Flow (NDF) (Sun et al., 2022) addresses these limitations by parameterising shape-conditioned deformations as neural ODE solutions, ensuring invertibility and topology preservation by construction. We extend NDF to the spatiotemporal domain, learning diffeomorphic flows that capture shape and motion variations across subjects and time, yielding a left ventricular atlas with dense, topology-preserving correspondences to individual cases. We validate on UK Biobank CMR data (Sudlow et al., 2015), demonstrating effective motion recovery

from sparse frames while preserving anatomical plausibility. Our contributions are: (1) extending NDF to 4D for cardiac motion modelling; (2) constructing an explicit spatiotemporal left ventricular atlas with topology-preserving correspondences; and (3) validating motion recovery on UK Biobank data with ground-truth evaluation.

## 2. Methods

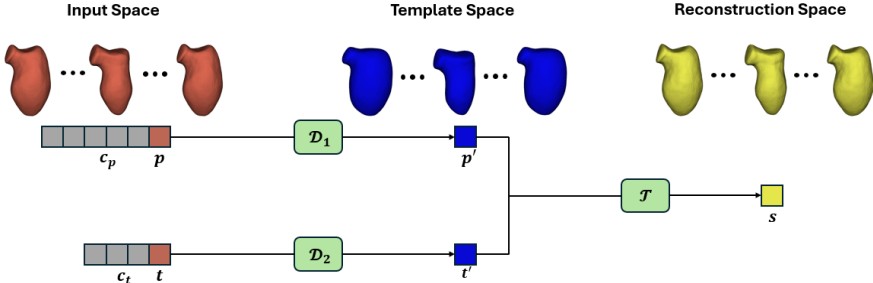

Figure 1: Overview of spatiotemporal NDF. A spatial warper $\mathcal{D}_1$ and temporal warper $\mathcal{D}_2$ map input spatial and temporal coordinates ($\mathbf{p}$, $t$) with subject-specific latent codes ($\mathbf{c}_p$, $\mathbf{c}_t$) to template coordinates ($\mathbf{p}'$, $t'$), which are passed to a single-shape DeepSDF (Park et al., 2019) $\mathcal{T}$ to predict signed distance $s$.

Rather than operating on 3D coordinates alone, we extend NDF to the spatiotemporal domain with two independent deformation modules for the spatial and temporal dimensions. Given a CMR sequence $X_i$, the network accepts ($\mathbf{p}$, $t$) and a subject-specific latent code $\mathbf{c}_i$ as input, where $t \in [0,1]$ denotes the normalised cardiac cycle phase and $\mathbf{p}$ denotes the 3D coordinates $(x, y, z)$. The signed distance field $\mathcal{F}$ becomes $\mathcal{F}(\mathbf{p}, t, \mathbf{c}_i) = \mathcal{T}(\mathcal{D}_1(\mathbf{p}, \mathbf{c}_{ip}), \mathcal{D}_2(t, \mathbf{c}_{it})) = \mathcal{T}(\mathbf{p}', t') = s$, where $s \in \mathbb{R}$ is the signed distance to the closest surface, $\mathcal{T} : \mathbb{R}^4 \to \mathbb{R}$ is the learned implicit atlas parameterised as a single-shape DeepSDF (Park et al., 2019), $\mathcal{D}_1 : \mathbb{R}^3 \times \mathbb{R}^m \to \mathbb{R}^3$ is a spatial diffeomorphic deformation module conditioned on spatial latent code $\mathbf{c}_{ip} \in \mathbb{R}^m$, and $\mathcal{D}_2 : \mathbb{R} \times \mathbb{R}^n \to \mathbb{R}$ is a temporal diffeomorphic deformation module conditioned on temporal latent code $\mathbf{c}_{it} \in \mathbb{R}^n$. Here $\mathbf{p}'$ and $t'$ denote the warped spatial and temporal coordinates in the template space, and $\mathbf{c}_i$ is decomposed into $\mathbf{c}_{ip}$ and $\mathbf{c}_{it}$ for the spatial and temporal deformations, respectively. The spatiotemporal atlas is learned implicitly during training without explicit supervision.

## 3. Experiments and Results

**Data Preprocessing**: We selected 100 subjects from the UK Biobank CMR database (80 training, 20 testing). CMR slices were segmented using nnU-Net (Isensee et al., 2021) and reconstructed into dense 3D volumes per frame via a shape completion pipeline (Xu et al., 2025). Left ventricular meshes were extracted via Marching Cubes (Lorensen and Cline, 1987) and processed following DeepSDF (Park et al., 2019) to sample 3D coordinates and compute signed distance values, with the first-frame bounding box applied across all frames to ensure temporal coherence. Each 50-frame sequence was made cyclic by duplicating

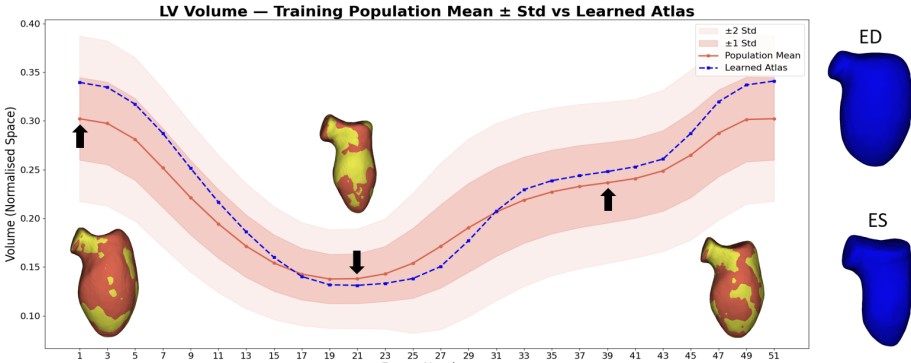

Figure 2: Volume curves of the left ventricle from the learned atlas compared to the training population, computed in the normalised space. ED and ES frames of the learned atlas are shown on the right (blue). Alignment between ground truth (red) and reconstructed meshes from the ED frame (yellow) is presented at different cardiac phases (indicated by black arrows): ED, ES, and diastasis.

the first frame as the last, then subsampled to 26 frames, with withheld frames serving as ground truth for motion recovery evaluation. Time coordinates were normalised to $[0, 1]$ and paired with their corresponding sampled 3D coordinates.

**Atlas Generation and Motion Recovery** Figure 2 shows the left ventricular volume curves computed in the learned template space compared to the training population average. The curves follow the expected contraction–relaxation pattern and differ from the population mean, indicating that the atlas captures meaningful cardiac dynamics. For motion recovery, we report chamfer distance (CD) and 95th percentile Hausdorff distance (HD95) under three settings: (1) 26 frames (same as training), (2) end-diastolic (ED) and end-systolic (ES) frames only, and (3) ED frame only (CT-style). The CD / HD95 results (mm, mean±std) are: $1.03 \pm 0.25$ / $2.40 \pm 0.68$ (26 frames), $1.24 \pm 0.57$ / $2.91 \pm 1.60$ (ED+ES), and $1.36 \pm 0.49$ / $3.25 \pm 1.34$ (ED only). Even with a single ED frame, the model achieves comparable accuracy, improving as more frames are provided at test time.

## 4. Conclusion and Future Work

We extend NDF to spatiotemporal atlas learning, demonstrating strong motion recovery on CMR sequences. Future work includes extending beyond the left ventricle to additional structures and pathological cases, and benchmarking against existing methods.

## Acknowledgments

YX and AAY acknowledge funding from EPSRC (EP/S022104/1, Z533762) and Wellcome/EPSRC (WT203148/Z/16/Z). YX acknowledges stipend funding from Heartflow Inc. MS and EP are employees of Heartflow Inc. This research has been conducted using the UK Biobank Resource under application number 88878.

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
