# OpenReview forum: "Spatiotemporal Neural Diffeomorphic Flow for Cardiac Atlas and Motion Recovery"
_MIDL.io/2026/Short_Papers — MIDL 2026 - Short Papers Poster_

### Official Review · Reviewer_2WdC · 2026-04-21
**Using spatiotemporal neural ODEs to learn the left ventricle atlas and deformation in a cardiac cycle**

**Rating:** 4
**Confidence:** 4

**Review:**

Overall, this is a good paper. It is clearly written. The novelty lies in using spatiotemporal neural ODEs to learn the left ventricle atlas and deformation in a cardiac cycle. The performance of atlas construction and left ventricle motion recovery is good. However, some details of the proposed architecture and baseline comparison were missing.

**Summary:**

This work proposes to use spatiotemporal neural ODEs to learn the left ventricle atlas and deformation in a cardiac cycle. The spatiotemporal neural ODEs can encode both spatial and temporal information from a cardiac image sequence. The method was evaluated on the UK Biobank CMR dataset with 100 subjects. The results show that the proposed method can learn the dynamic atlas and recover missing left ventricle shapes in a cardiac cycle.

**Strengths:**

1. Using spatiotemporal neural ODEs to learn the left ventricle atlas and deformation in a cardiac cycle is a novel idea.
2. And the performance of atlas construction and left ventricle motion recovery is good.

**Weaknesses:**

1. It is unclear how to construct the final dynamic atlas from the template space. Did you calculate the mean of the training datapoints as the atlas?
2. How to get the subject-specific codes?
3. No baselines were included for comparison.

**Justification Of Rating:**

The idea is novel and the performance looks good. However, some details and baseline comparison were missing.

---

### Decision · Program_Chairs · 2026-05-08

Accept (Poster)